

# Out-of-equilibrium phase transitions induced by Floquet resonances in a periodically quench-driven XY spin chain

Sergio Enrique Tapias Arze[1], Pieter W. Claeys[2],
Isaac Pérez Castillo[3,4*] and Jean-Sébastien Caux[1]

**1** Institute for Theoretical Physics Amsterdam and Delta Institute for Theoretical Physics,
University of Amsterdam, Science Park 904, 1098 XH Amsterdam, The Netherlands
**2** TCM Group, Cavendish Laboratory, University of Cambridge, Cambridge, UK
**3** Departamento de Física Cuántica y Fotónica, Instituto de Física, UNAM,
P.O. Box 20-364, 01000 Mexico Distrito Federal, Mexico
**4** London Mathematical Laboratory, 8 Margravine Gardens,
London, W6 8RH, United Kingdom

⋆ isaacpc@fisica.unam.mx

## Abstract

We consider the dynamics of an XY spin chain subjected to an external transverse field which is periodically quenched between two values. By deriving an exact expression of the Floquet Hamiltonian for this out-of-equilibrium protocol with arbitrary driving frequencies, we show how, after an unfolding of the Floquet spectrum, the parameter space of the system is characterized by alternations between local and non-local regions, corresponding respectively to the absence and presence of Floquet resonances. The boundary lines between regions are obtained analytically from avoided crossings in the Floquet quasi-energies and are observable as phase transitions in the synchronized state. The transient behaviour of dynamical averages of local observables similarly undergoes a transition, showing either a rapid convergence towards the synchronized state in the local regime, or a rather slow one exhibiting persistent oscillations in the non-local regime, where explicit decay coefficients are presented.

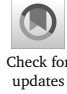

# 1 Introduction

The study of out-of-equilibrium systems has recently moved to the forefront of research in many-body physics, motivated largely by the desire to create interesting states of matter with properties beyond those achievable in equilibrium. In periodically-driven systems, which exist in nature but can also be created in the laboratory, driving can be used as a probe or to generate non-trivial phases [1–3]. Floquet theory [4] is then used to describe these situations. Whereas all relevant properties of the periodic dynamics are encoded in the Floquet Hamiltonian, finding exact solutions remains a severe challenge, and these are usually limited to few-body systems. This poses severe limitations to the application of Floquet theory when a large number of components are involved, as in condensed matter physics. However, in free-fermion models and related spin chain systems, the Floquet Hamiltonian sometimes takes a simpler form due to their underlying algebraic structure [5]. These models can then shed light on periodically-driven systems, while describing experimentally-relevant situations [6–8], and being relevant in various fields of physics [1, 9, 10]. Previous works have explored some of these scenarios [11, 12], and numerical studies have also been performed for non-integrable systems [13–16], where many-body resonances are of major interest [17–23].

Yet, exact expressions for Floquet Hamiltonians at arbitrary driving frequencies and explicit connections between the dynamics and the quasi-energy spectrum, from which much could be learned on equilibration and synchronization in driven systems, are lacking in most cases. In this paper, we consider an antiferromagnetic spin-1/2 XY chain subject to periodic quenches of a transverse magnetic field. Although the equilibrium properties of this model are well-known [24–26], and despite studies of its out-of-equilibrium behaviour in certain scenarios (e.g., [27–33]), an exact description of its Floquet dynamics valid in the entire parameter space is currently missing. The XY and related models have been fundamental for understanding quantum phase transitions [32, 34, 35] and quantum quenches [36, 37], and can certainly play a similar role for the physics of periodic driving. Its Floquet Hamiltonian is obtained exactly, and we investigate its local/non-local nature for different driving parameters. A link is established between the locality of the Floquet Hamiltonian and the presence of avoided crossings between its eigenstates, which can be interpreted as Floquet resonances. These resonances are generally not observable in the thermodynamic limit, but the free-fermionic nature of the model makes them visible in the single-particle spectrum, leading to a slow convergence of observables towards a synchronized state (i.e., the steady state of the stroboscopic time evo-

lution [11,12]) with persistent oscillations in the non-local regime. These results are expected to be generic for any system mappable to a free fermion model. This study also allows to naturally connect Floquet resonances [18] with the appearance of long-range correlations as presented in Refs. [38,39]. The phenomena analyzed are also expected to be observable in string order parameters, characterizing similar transitions in the *Floquet ground state* of driven Ising models [40,41].

This paper is organised as follows. In Sec. 2, we start by providing a brief overview of Floquet theory. This is subsequently applied to a periodically driven XY spin chain, whose corresponding Floquet Hamiltonian is explicitly derived (see Secs. 3 and 4) and its effective interactions thoroughly discussed (see Sec. 5). In Sec. 6, a detailed account is presented on the correspondence between the locality, or lack thereof, of the Floquet Hamiltonian and the dynamics of observables. The last section is reserved for conclusions and possible future avenues of research.

## 2 Floquet theory

Analogously to the Bloch theorem for crystalline solids, given a time periodic Hamiltonian $\hat{H}(t) = \hat{H}(t + T)$ with $T = 2\pi/\omega_0$, one can construct a complete set of solutions of the Schrödinger equation, the so-called Floquet states [1, 4, 11], which are periodic in time up to a phase factor,

$$|\Psi_\alpha(t)\rangle = e^{-i\epsilon_\alpha^{[T]}t} |\Phi_\alpha(t)\rangle . \tag{1}$$

Here the states $|\Phi_\alpha(t)\rangle$ are called Floquet modes and are periodic $|\Phi_\alpha(t + T)\rangle = |\Phi_\alpha(t)\rangle$, with $\alpha$ representing a set of quantum numbers. The real quantities $\epsilon_\alpha^{[T]}$ are known as the Floquet quasienergies. Due to the periodicity, these quasienergies can be translated to the first Brillouin zone $[-\pi/T, \pi/T]$, just like the quasimomenta of Bloch waves.

To derive expressions for the Floquet modes and quasienergies, we first notice that, by definition, a Floquet state obeys the Schrödinger equation

$$\hat{U}(t, t_0) |\Psi_\alpha(t_0)\rangle = |\Psi_\alpha(t)\rangle , \tag{2}$$

with evolution operator $\hat{U}(t_2, t_1) = \mathcal{T}_t \exp\left[-i \int_{t_1}^{t_2} \hat{H}(t)dt\right]$. One can easily show that, for time-periodic Hamiltonians, this evolution operator obeys that $U(t_0 + nT, t_0) = U(t_0 + T, t_0)^n$. In terms of the Floquet modes the preceding evolution equation becomes

$$\hat{U}(t, t_0) |\Phi_\alpha(t_0)\rangle = e^{-i\epsilon_\alpha^{[T]}(t-t_0)} |\Phi_\alpha(t)\rangle . \tag{3}$$

Eq. (3) tells us, on the one hand, how Floquet modes evolve and, on the other, that for stroboscopic times $t = t_0 + nT$, the Floquet modes are eigenfunctions of the evolution operator $\hat{U}(t_0 + nT, t_0)$, that is

$$\hat{U}(t_0 + nT, t_0) |\Phi_\alpha(t_0)\rangle = e^{-i\epsilon_\alpha^{[T]}nT} |\Phi_\alpha(t_0)\rangle . \tag{4}$$

Moreover, since a unitary operator can always be written as the exponential of a Hermitian operator, this allows us to introduce the so-called Floquet Hamiltonian $\hat{H}_F^{[t_0,T]}$ so that $\hat{U}(t_0 + T, t_0) = e^{-i\hat{H}_F^{[t_0,T]}T}$. The Floquet Hamiltonian is time-independent, but depends both on the driving period, as well as on the arbitrary choice of the initial time. Since Floquet Hamiltonians for different initial times are related by a gauge transformation, $t_0$ is referred to as

the "Floquet gauge". Thus, the properties of the system at stroboscopic times $t_0 + nT$ can be determined by studying the static Floquet Hamiltonian.

Finally, it can be shown that the evolution of a Floquet mode $\left|\Phi_\alpha^{[T]}(t_0)\right\rangle \rightarrow \left|\Phi_\alpha^{[T]}(t)\right\rangle$ (where the superscript $T$ now explicitly shows its dependence on the period) from $t_0$ to $t = t_0 + nT + \delta t$ is given by

$$\left|\Phi_\alpha^{[T]}(t)\right\rangle = e^{i\epsilon_\alpha^{[T]}\delta t}\hat{U}(t_0 + \delta t, t_0)\left|\Phi_\alpha^{[T]}(t_0)\right\rangle. \tag{5}$$

## 3  Driving the XY spin chain

In the following, Floquet theory will be applied to the XY spin chain, where we pay special attention to the structure of the Floquet Hamiltonian. The instantaneous XY Hamiltonian is given by

$$H^{\mathrm{XY}} = J\sum_{j=1}^{N}\left[(1+\gamma)S_j^x S_{j+1}^x + (1-\gamma)S_j^y S_{j+1}^y - \frac{h(t)}{J}S_j^z\right], \tag{6}$$

with superexchange parameter $J > 0$, anisotropy $\gamma \in [0,1]$, and $h(t)$ a time-periodic external magnetic field with non-negative average value. The spin-1/2 operators $S_j^\alpha$, $\alpha \in \{x, y, z\}$, are taken to obey periodic boundary conditions. Performing a Jordan-Wigner transformation and a Fourier transform to spinless fermion operators $\{c_k^\dagger, c_k\}$, and taking care of the boundary conditions [24] (see Appendix A for details of the derivation), this instantaneous Hamiltonian can be mapped to

$$H^{\mathrm{XY}} = -J\sum_{k>0}\left[(\cos(k) - h(t)/J)(c_k^\dagger c_k + c_{-k}^\dagger c_{-k}) + i\gamma\sin(k)(c_k^\dagger c_{-k}^\dagger - c_{-k}c_k)\right] - \frac{h(t)N}{2}, \tag{7}$$

which (assuming even filling in subsectors $\{k, -k\}$) is a direct sum of operators acting on two-dimensional subspaces spanned by $\{|0\rangle, c_k^\dagger c_{-k}^\dagger|0\rangle\}$. Observing the system stroboscopically at times $t_0 + nT$ for $n \in \mathbb{N}$, the evolution operator can be written as

$$U(t_0 + nT, t_0) = e^{-iH_{\mathrm{F}}^{[t_0,T]}nT}. \tag{8}$$

The so-called Floquet Hamiltonian $H_{\mathrm{F}}^{[t_0,T]}$ can be obtained by diagonalizing the evolution operator for a single driving cycle. In each two-dimensional momentum subspace, this operator can be represented as

$$U_k(t_0 + T, t_0) = u_{k,0}\mathbb{1}_k + i\sum_{\alpha\in\{x,y,z\}} u_{k,\alpha}\sigma_k^\alpha, \tag{9}$$

with Pauli matrices $\sigma_k^\alpha$. Solving the eigenvalue problem provides the Floquet modes and the quasienergies. After some algebra one obtains the Floquet modes

$$\left|\Phi_{k,\sigma}^{[T]}(t_0)\right\rangle = b_\sigma \begin{pmatrix} 1 \\ \sigma\frac{u_{k,x}+iu_{k,y}}{\sqrt{1-u_{k,0}^2}+\sigma u_{k,z}} \end{pmatrix}, \tag{10}$$

with normalization factor

$$b_\sigma = \left[\frac{\sqrt{(1-u_{k,0}^2)}+\sigma u_{k,z}}{2\sqrt{1-u_{k,0}^2}}\right]^{\frac{1}{2}}, \tag{11}$$

and corresponding eigenvalues $\lambda_\sigma = e^{-i\sigma\epsilon_T(k)T}$ with $\sigma = \pm 1$. Here $\epsilon_T(k)$ are the quasi-energies given by

$$\epsilon_T(k) = -\frac{1}{T}\arccos(u_{k,0}). \tag{12}$$

This allows us to write the following pseudospin representation of the Floquet Hamiltonian (up to an overall constant)

$$H_F^{[t_0,T]} \equiv \sum_{k>0} \epsilon_T(k)\mathbf{n}_k \cdot \sigma_k, \tag{13}$$

with

$$\mathbf{n}_k \equiv \left( \frac{u_{k,x}}{\sqrt{1-u_{k,0}^2}}, \frac{u_{k,y}}{\sqrt{1-u_{k,0}^2}}, \frac{u_{k,z}}{\sqrt{1-u_{k,0}^2}} \right). \tag{14}$$

It is, however, more illuminating to write the resulting Floquet Hamiltonian in terms of the Jordan-Wigner fermions in real space, leading to the following result:

$$H_F^{[t_0,T]} = \sum_{j=1}^{N} \sum_{\ell=1}^{(N-1)/2} \left[ \mathcal{A}_\ell c_j^\dagger c_{j+\ell} + \mathcal{B}_\ell c_j^\dagger c_{j+\ell}^\dagger + \text{h.c.} \right] + \mathcal{A}_0 \sum_{j=1}^{N} c_j^\dagger c_j, \tag{15}$$

where we can see that the driving may induce non-local interactions $\mathcal{A}_\ell$ and $\mathcal{B}_\ell$.

So far the results are general, as they only rely on the algebraic structure of the system. The problem is now that obtaining explicitly exact, analytical expressions for Euler parameters $u_{k,\alpha}$ is not a trivial matter. One protocol that allows for such solutions consists of periodically quenching between two values of the time dependent external field $h(t)$,

$$h(t) = \begin{cases} h_1, & 0 \le t < \alpha T \mod T \\ h_2, & \alpha T \le t < T \mod T \end{cases}. \tag{16}$$

In this case, one can use the algebra of the Pauli matrices to determine the effective parameters $u_{k,\alpha}$. In the following, we take $t_0 = 0$ for simplicity. For this Floquet gauge, we obtain:

$$\begin{aligned}
u_{k,0} &= \cos(E_1(k)\alpha T)\cos(E_2(k)(1-\alpha)T) \\
&\quad - \sin(E_1(k)\alpha T)\sin(E_2(k)(1-\alpha)T)\cos(\Delta_k), \\
u_{k,x} &= \sin(E_1(k)\alpha T)\sin(E_2(k)(1-\alpha)T)\sin(\Delta_k), \\
u_{k,y} &= \cos(E_1(k)\alpha T)\sin(E_2(k)(1-\alpha)T)\sin(\theta_{2,k}), \\
&\quad + \cos(E_2(k)(1-\alpha)T)\sin(E_1(k)\alpha T)\sin(\theta_{1,k}), \\
u_{k,z} &= -\cos(E_1(k)\alpha T)\sin(E_2(k)(1-\alpha)T)\cos(\theta_{2,k}) \\
&\quad - \cos(E_2(k)(1-\alpha)T)\sin(E_1(k)\alpha T)\cos(\theta_{1,k}),
\end{aligned}$$

where $E_i(k)$ are the eigenvalues of the XY Hamiltonian with field $h_i$, $\Delta_k = \theta_{2,k} - \theta_{1,k}$, and $\theta_{i,k}$ is the corresponding Bogoliubov angle for that same external field. This is defined as $\sin(\theta_{i,k}) = -\lambda_k/\sqrt{\xi_{i,k}^2 + \lambda_k^2}$, with $\lambda_k = \gamma\sin(k)$ and $\xi_{i,k} = \cos(k) - h_i/J$. Detailed expressions of the parameters $u_{k,\alpha}$ for arbitrary Floquet gauge choices can be found in Appendix B.

Before analyzing the influence of these possibly non-local interactions, let us discuss their origin by considering the Floquet-Magnus expansion [13, 42]. In the infinite driving frequency limit $T \to 0$, the Floquet Hamiltonian reduces to the time-averaged Hamiltonian $H_{av}$. Eigenstates of $H_{av}$ whose single-particle energy difference (within a single momentum sector) equals

an integer multiple of the driving frequency $2\pi/T$ are said to be quasi-degenerate or resonant and, in the same way that a small perturbation on a stationary Hamiltonian can strongly couple degenerate eigenstates of the unperturbed Hamiltonian, deviations of $e^{-iH_F^{[t_0,T]}T}$ from $e^{-iH_{av}T}$ can strongly couple quasi-degenerate eigenstates [17–23]. These so-called *Floquet resonances* give rise to avoided crossings that result into the Floquet eigenstates being linear combinations of two such (near-) resonant states. As this mechanism cannot be represented by local operators, the Floquet Hamiltonian contains long-range interactions if resonances are present (note that other mechanisms can also lead to non-local terms in certain circumstances [43, 44]). In such situations, the Magnus expansion is not expected to converge [13, 45–48], hence exact solutions are important for the description of these resonances.

In Fig. 1 we show the quasi-energy spectrum as a function of $T$, where the intervals within which avoided crossings occur can be clearly distinguished. Since quasi-energies are only defined up to shifts $2\pi m/T, m \in \mathbb{N}$, in Fig. 1 the quasi-energies are restricted to the first Brillouin Zone $[-\pi/T, \pi/T]$, which is known as the 'folding' of the spectrum [13].

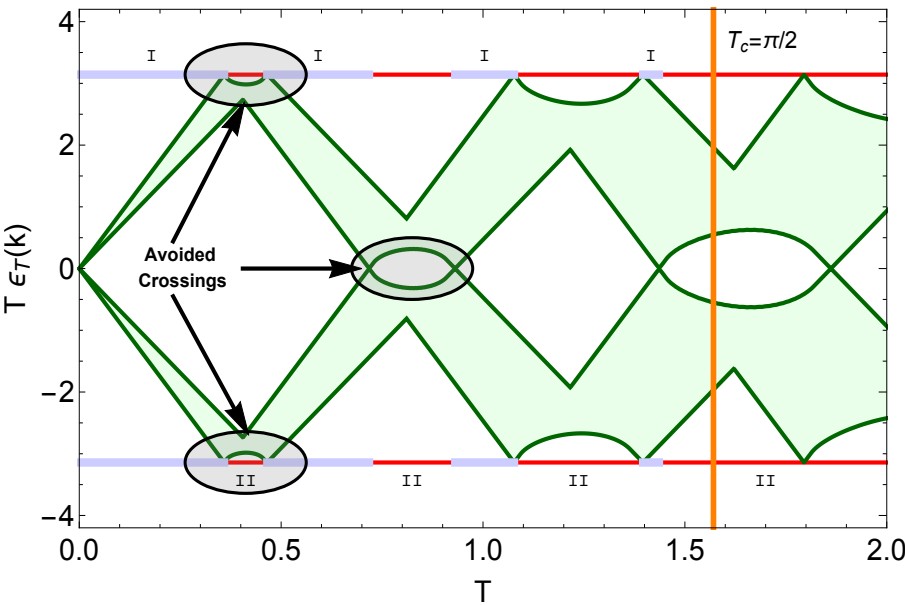

Figure 1: Continuous quasi-energy spectrum (in pale green) as a function of the period in the first Brillouin zone. Avoided crossings are present in the red intervals, indicated as II, and absent in the pale blue intervals, denoted as I. The vertical orange line marks the value of $T$ above which no more intervals of type I are found. In this figure, we have used the following values of the parameters: $J = 1$, $\gamma = 0.8$, $h_1 = 15$, $h_2 = 0.5$, $\alpha = 0.5$, and $N = 10001$.

# 4  Unfolding of the quasi-energy spectrum

By construction the Floquet Hamiltonian possesses the same eigenstates as the evolution operator, and its eigenvalues $\pm\epsilon_T(k) = \mp\arccos(u_{k,0})/T$ follow from the logarithm of the corresponding eigenvalues of $U_k(t_0 + T, t_0)$. As such, these quasi-energies are defined up to shifts $2\pi m/T$, with $m \in \mathbb{N}$, whose branch choice is seen as a discrete gauge freedom [49]. While the gauge choice for the quasi-energies is immaterial for the time evolution of the system, leading to a Floquet Hamiltonian that is similarly only defined up to a gauge freedom, its effects on

the latter are, however, much more involved. As will be argued in the following, this folding might introduce unphysical interactions in the local Hamiltonian, and the spectrum needs to be explicitly unfolded in order to highlight the effects of Floquet resonances.

Although there exists no general way of doing so [18], in the current model we have the advantage that, due to the free-fermionic structure, the Floquet Hamiltonian factorises into two-dimensional subspaces and has a symmetric spectrum. Therefore, the single-particle quasi-energy spectrum can be explicitly unfolded by either adding or subtracting the corresponding multiple of $2\pi/T$ whenever a resonance condition is met (see Fig. 2 for a graphical representation of the unfolding).

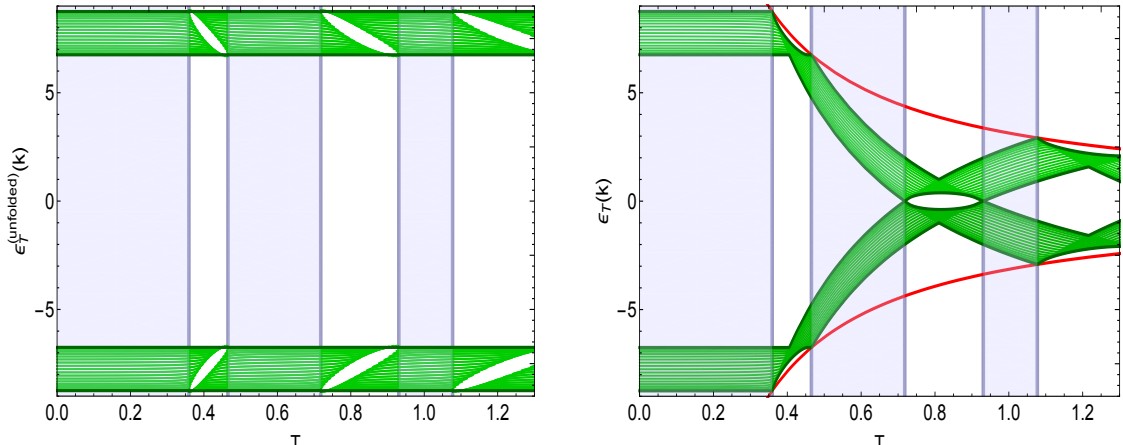

Figure 2: Unfolded (left) and folded (right) quasienergies $\epsilon_T(k)$ at different values of the driving period $T$ for a small system with $N = 41$. In all local regions the unfolding returns a spectrum close to that of the time-averaged Hamiltonian, whereas the avoided crossings in the non-local regions are reflected in opening gaps in the unfolded Floquet spectrum. The edges of the Brillouin zone $\pm\pi/T$ are marked by dashed red lines in the right figure. The system parameters correspond to $\alpha = 1/2$, $\gamma = 0.5$, $J = 1$, $h_1 = 1/2$, $h_2 = 15$.

Here, we present two equivalent ways of explicitly unfolding the Floquet spectrum. The first one takes advantage of the symmetry of the spectrum of the time-averaged and Floquet Hamiltonians, which implies that an avoided crossing takes place whenever the value of a quasienergy approaches the edge of the Brillouin zone. This is precisely the mechanism that keeps the spectrum found in (12) inside the interval $[-\pi/T, \pi/T]$. For a given value of $k$ and as the period increases, the quasi-energy $\epsilon_T(k)$ will have a first resonance at $T = \pi/E^{av}(k)$. For periods above this value, we can unfold the quasienergy out of the first Brillouin zone by shifting it by $2\pi/T$. The spectrum will now be confined inside the second Brillouin zone, and in order to further unfold it we need a shift of $-2 \times 2\pi/T$ for periods larger than $2\pi/E^{av}(k)$, where a second resonance takes place. By repeating this process iteratively, the full unfolding can be written as

$$
\begin{aligned}
\epsilon_T^{(\text{unfolded})}(k) &= \epsilon_T(k) - \sum_{n=1}^{\infty}(-1)^n\Theta(TE^{av}(k)-n\pi)\frac{2\pi n}{T} \\
&= \epsilon_T(k) - \frac{2\pi}{T}\sum_{n=1}^{\lfloor TE^{av}(k)/\pi\rfloor}(-1)^n n\,,
\end{aligned}
\tag{17}
$$

with $\Theta(x)$ the Heaviside function.

An alternative way of unfolding the spectrum is by making the connection with the $\gamma \to 0$ limit of driven XX models. In this limit, the driving becomes trivial since the XX Hamiltonians commute at all values of $h$ and $J$, leading to a Floquet operator whose eigenstates are exactly those of the time-averaged Hamiltonian $H_{av}$. The spectrum can now be unfolded by demanding that the Floquet Hamiltonian reduces to $H_{av}$ at all values of the driving period $T$ if we take this limit. Namely, for $\gamma = 0$ the evolution operator in the pseudo-spin representation is

$$U_k(t_0 + T, t_0) = \cos(E^{av}(k)T)\mathbb{1} - i \sin(E^{av}(k)T)\sigma_k^z, \tag{18}$$

with $E^{av}(k) = |J\cos(k) - h_{av}|$ and $h_{av} = \alpha h_1 + (1-\alpha)h_2$, while the Floquet Hamiltonian reduces to

$$
\begin{aligned}
H_F &= \sum_{k>0} \frac{1}{T} \arccos\left[\cos(E^{av}(k)T)\right] \operatorname{sgn}\left[\sin(E^{av}(k)T)\right] \sigma_k^z + \text{Cst.} \\
&= \sum_{k>0} \left( E^{av}(k) - \left\lfloor \frac{E^{av}(k)T + \pi}{2\pi} \right\rfloor \frac{2\pi}{T} \right) \sigma_k^z + \text{Cst.}
\end{aligned}
\tag{19}
$$

It is clear that this Hamiltonian shares the eigenstates of the time-averaged Hamiltonian, where the eigenvalues have acquired an appropriate shift compared to $E^{av}$ because of the term $\arccos\left[\cos(E^{av}(k)T)\right]/T$. The shifts introduced by the arccos function can be explicitly undone by defining the unfolded quasienergies as

$$\epsilon_T^{(\text{unfolded})}(k) = \epsilon_T(k) + \operatorname{sign}\left[\sin(E^{av}(k)T)\right] \left\lfloor \frac{E^{av}(k)T + \pi}{2\pi} \right\rfloor \frac{2\pi}{T}. \tag{20}$$

In the limit $\gamma \to 0$, at each value of the driving period the Floquet Hamiltonian now reduces to $H_{av}$, which has a guaranteed unfolded spectrum. It is easy to show that Eqs. (17) and (20) are equivalent. While the necessary shifts depend on $E^{av}(k)$, this only determines the edges of the shifts due to the sign- and floor-function, similar as in the determination of the boundaries of the different regions. The resulting shifts are introduced at the avoided crossings, and as such the resulting Floquet Hamiltonian will showcase the different dynamical behaviours in the different regions.

With this choice of unfolding one can check that we recover the expected results for various limits of the parameters. For instance, if $\gamma = 0$ all instantaneous Hamiltonians commute and the driving trivializes, leading to $H_F^{[t_0,T]} = H_{av}$ at all driving frequencies, as returned by the proposed unfolding. Another example is the static case, which can be achieved by either $h_1 = h_2$, $\alpha = 0$ or $\alpha = 1$, for which the Floquet Hamiltonian reduces to the static one.

## 5 Induced interactions in the Floquet Hamiltonian

Now that the spectrum has been unfolded, we can return to the couplings $\mathcal{A}_\ell$ and $\mathcal{B}_\ell$ in the Floquet Hamiltonian given by Eq. (15). In particular, we are interested in whether or not the driving introduces significant couplings at ranges larger than the nearest-neighbour interactions present in the instantaneous Hamiltonian. With our choice of driving protocol, these couplings are given by

$$
\begin{aligned}
\mathcal{A}_\ell &= \frac{2}{N} \sum_{k>0} \frac{u_{k,z}}{\sqrt{1 - u_{k,0}^2}} \cos(\ell k) \epsilon_T^{(\text{unfolded})}(k), \\
\mathcal{B}_\ell &= -\frac{2}{N} \sum_{k>0} \frac{u_{k,y} + iu_{k,x}}{\sqrt{1 - u_{k,0}^2}} \sin(\ell k) \epsilon_T^{(\text{unfolded})}(k).
\end{aligned}
\tag{21}
$$

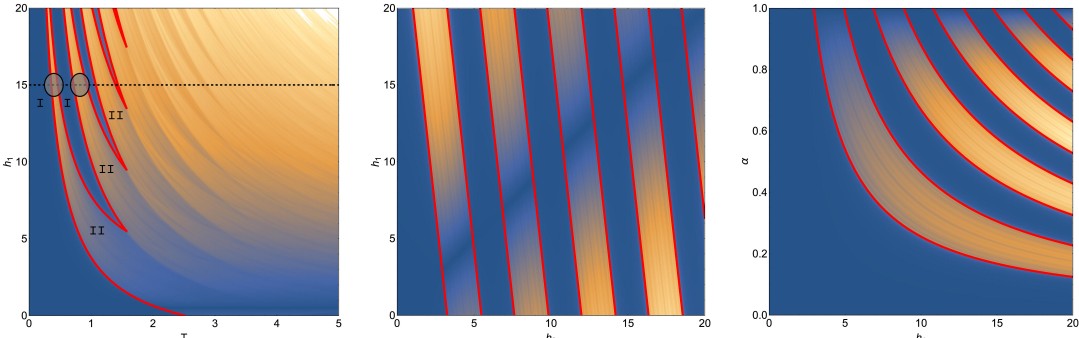

Figure 3: Density plots of the measure $\rho$ as a proxy of the non-locality of the Floquet Hamiltonian in the $(T, h_1)$- , $(h_1, h_2)$-, and $(h_1, \alpha)$-planes (from left to right). In all three figures, $J = 1$, $L = 10$, and $N = 10001$. The other parameters are $\gamma = 0.8$, $h_2 = 0.5$, $\alpha = 0.5$ (left figure); $\gamma = 0.2$, $\alpha = 0.1$, $T = 0.8$ (middle figure); and $\gamma = 0.2$, $h_2 = 0.5$, $T = 0.8$ (right figure). The red lines separating the different regions are the predictions according to Eqs. (27). Note the local line $h_1 = h_2$ in the middle plot, which corresponds to a trivial quench-driven protocol for which the Floquet Hamiltonian is precisely the average one.

To visualise the non-locality of the the Floquet Hamiltonian in the parameter space we introduce the measure

$$\rho = \frac{1}{2(L-1)} \sum_{\ell \geq 2}^{L} \left\{ |\mathcal{A}_\ell| + \sqrt{||\mathcal{B}_\ell||^2} \right\} . \tag{22}$$

This quantity measures the average strength of interactions up to range $L$, excluding those of range 1 as well as the effective on-site potential. Therefore, it allows us to probe only interaction ranges which were not present in the static Hamiltonians and, as such, the non-local long-range behaviour of the Floquet Hamiltonian. As mentioned before, as $T$ goes to zero, one retrieves the time-averaged Hamiltonian, and $\rho = 0$. When $T$ increases, one would naively expect an enhancement of longer range interactions, reflected in an increase of $\rho$. However, this is not always the case. As can be seen in Fig. 3, which shows density plots of $\rho$ in the planes $(T, h_1)$, $(h_1, h_2)$, and $(h_1, \alpha)$, alternating regions where interactions are either local (blue regions corresponding to $\rho \simeq 0$) or non-local (orange regions corresponding to $\rho > 0$) can be clearly distinguished.

Furthermore, in Fig. 4 we show the amplitudes of the couplings $\mathcal{A}_\ell$ (n. b. $\mathcal{B}_\ell$-couplings behave in an analogous manner) along the dotted line $h_1 = 15$ corresponding to the left plot in Fig. 3, where it can be seen that inside the local regime (highlighted by the pale blue background) only the local couplings are non-neglibible, whereas in the non-local regime all the couplings play a role in the Floquet Hamiltonian. These local to non-local transitions are induced by the appearance of avoided crossings, as indicated in Fig. 1, since the latter are the origin of the long-range interactions. As such, the boundary lines separating these two kinds of regions, drawn as red lines in Fig. 3, can be analytically derived by looking for these resonance conditions.

To do so, we start by noticing that, due to the symmetry of the spectrum of the time-averaged Hamiltonian, resonances begin to occur whenever the quasienergies get close to the first Brillouin zone (as can also be observed from Fig. 1). If we were to start at small values of $T$ we expect the quasienergies to correspond to the eigenvalues of the time-averaged Hamiltonian,

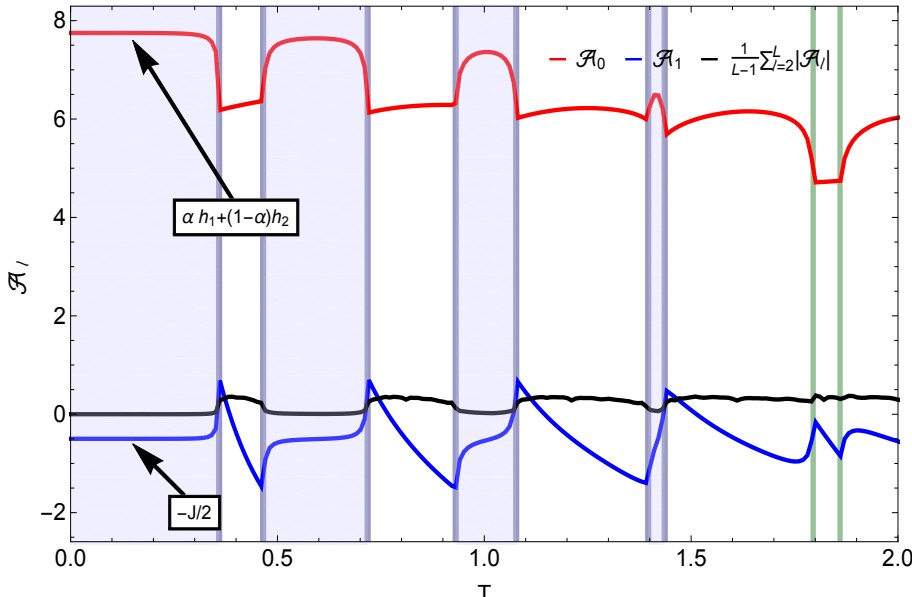

Figure 4: Effective couplings $\mathcal{A}_\ell$ as a function of the period. The red line shows the effective on-site magnetic field $\mathcal{A}_0$, the blue line the effective nearest neighbours interaction $\mathcal{A}_1$, and the black one the average strength of the remaining effective couplings. The latter becomes negligible inside the local regions indicated by the pale blue background.

given by

$$E_\sigma^{av}(k) = \sigma J \sqrt{\gamma^2 \sin^2(k) + (\cos(k) - h_{av}/J)^2}, \tag{23}$$

with $h_{av} = \alpha h_1 + (1 - \alpha)h_2$ the average magnetic field, which we take to be non-negative without loss of generality. This dispersion has its maximum for $k = \pi$, so we expect that this would be the first momentum mode to reach the boundary of the Brillouin zone. The last mode to reach it corresponds with the minimum $k^\star$ of the dispersion, given by

$$k^\star = \begin{cases} 0 & \text{if} \quad h_{av} \geq J(1 - \gamma^2), \\ \arccos(h_{av}/J(1 - \gamma^2)) & \text{if} \quad h_{av} < J(1 - \gamma^2). \end{cases} \tag{24}$$

This will define bands within which there are avoided crossings, delimited by the conditions

$$\Delta E^{av}(\pi) \equiv E_+^{av}(\pi) - E_-^{av}(\pi) = m \frac{2\pi}{T},$$
$$\Delta E^{av}(k^\star) \equiv E_+^{av}(k^\star) - E_-^{av}(k^\star) = m \frac{2\pi}{T}, \tag{25}$$

with $m \in \mathbb{N}$, giving the following equations

$$2(J + h_{av}) = m \frac{2\pi}{T}, \tag{26}$$

and

$$\begin{cases} 2|J - h_{av}| = m \frac{2\pi}{T} & \text{if} \quad h_{av} \geq J(1 - \gamma^2), \\ 2\gamma \sqrt{J^2 - \frac{h_{av}^2}{1 - \gamma^2}} = m \frac{2\pi}{T} & \text{if} \quad h_{av} < J(1 - \gamma^2). \end{cases} \tag{27}$$

We can now use the set of equations (27) to find the boundary lines that appear in the density plots of Fig. 3. Consider, for instance, the left plot of Fig. 3 in the $h_{av} \geq J(1-\gamma^2)$ region, and let $T_\pi^{(m)}$ and $T_0^{(m)}$ denote the values of the period $T$ at which quasienergies at $k = \pi$ and $k = 0$ are close to the edge of the first Brillouin zone. These two values are such that $T_\pi^{(m)} \leq T_0^{(m)}$. Therefore, along the $T$ axis, avoided crossings are roughly within the set of disjoint intervals $\cup_{m \geq 0}[T_\pi^{(m)}, T_0^{(m)}]$, where the Floquet Hamiltonian is non-local. Alternatively, in the complementary set of disjoint intervals $\cup_{m \geq 0}[T_0^{(m)}, T_\pi^{(m+1)}]$, the Floquet Hamiltonian is roughly local. Looking precisely at the left panel of Fig. 3, we notice that the region of local interactions disappears above a certain value of $T$. This obviously must correspond to $T_0^{(m)} = T_\pi^{(m+1)} = T_c$. Solving this equation in the region where $h_{av} \geq J$ gives

$$
\begin{cases} J + h_c = \dfrac{\pi(m+1)}{T_c} \\ J - h_c = \dfrac{\pi m}{T_c} \end{cases} \Rightarrow \begin{cases} T_c = \dfrac{\pi}{2J} \\ h_c = J(2m+1) \end{cases}, \tag{28}
$$

where the set of $h_c$ is the family of values of average magnetic field where two boundaries intersect. In particular, in the left panel of Fig. 3 the end points of the non-local region in the $(T, h_1)$-plane correspond to the collection of points $(T_c, h_{1c}) = (\pi/2, 2(2m+1) - 1/2)$ for $m = 1, 2, 3, \ldots$. In the same way, one can derive the corresponding expressions for these intersections in the regimes $h_{av} < J(1-\gamma^2)$ and $J(1-\gamma^2) \leq h_{av} < J$.

To form a more explicit criterion for the locality of the interactions, we can look at the asymptotic decay of the couplings with increasing distance. We consider the couplings to behave as

$$
\mathcal{A}_\ell \sim f(\ell) + \frac{g(\ell)}{N}, \tag{29}
$$

for large $\ell$ and $N$. Here, $f(\ell)$ is the dominant decaying term in the thermodynamic limit and $g(\ell)$ the leading finite size correction to this decay. Let us first find it in the non-local regions, where the arguments of the sums in Eq. (21) present discontinuities due to the unfolding of the quasi-energy spectrum at points $\{k_i\}_{i=1}^n$ for which the Floquet states are resonant. In the limit of large $N$ all summations can be exchanged by integration, and defining $a_k = \epsilon_T(k) u_{k,z} / \sqrt{1 - u_{k,0}^2}$ and integrating by parts then returns

$$
\lim_{N \to \infty} \mathcal{A}_\ell = \frac{2}{2\pi} \text{Re} \left[ \int_0^{k_1} a_k e^{ik\ell} dk + \sum_{i=2}^{n-1} \int_{k_i}^{k_{i+1}} a_k e^{ik\ell} dk + \int_{k_n}^{\pi} a_k e^{ik\ell} dk \right]
$$
$$
= \frac{1}{\pi\ell} \sum_{i=1}^n (a_{k_i}^- - a_{k_i}^+) \sin(k_i \ell) + O(\ell^{-2}),
$$

with $a_{k_i}^\pm = \lim_{k \to k_i^\pm} a_k$. Thus, the couplings decay as $f(\ell) \propto \ell^{-1}$ in the non-local regions, as can be seen in Fig. 5. The deviations from the predicted decay for large $\ell$ are due to finite size corrections, which can similarly be obtained, as shown in Appendix C.

As for the local regions, there are no discontinuities in the integrand, and further analysis is necessary:

$$
\lim_{N \to \infty} \mathcal{A}_\ell = \frac{1}{\pi} \text{Re} \int_0^\pi a_k e^{ik\ell} dk = \frac{1}{\pi} \text{Re} \left[ \frac{1}{i\ell} \left[ \frac{da_k}{dk} e^{ik\ell} \right]_0^\pi - \frac{1}{i\ell} \int_0^\pi \frac{da_k}{dk} e^{ik\ell} dk \right]
$$
$$
= \frac{1}{\pi} \sum_{n \, \text{odd}} \frac{(-1)^{(n+1)/2}}{\ell^{n+1}} \left[ (-1)^\ell \frac{d^n a_k}{dk^n} \bigg|_\pi - \frac{d^n a_k}{dk^n} \bigg|_0 \right]. \tag{30}
$$

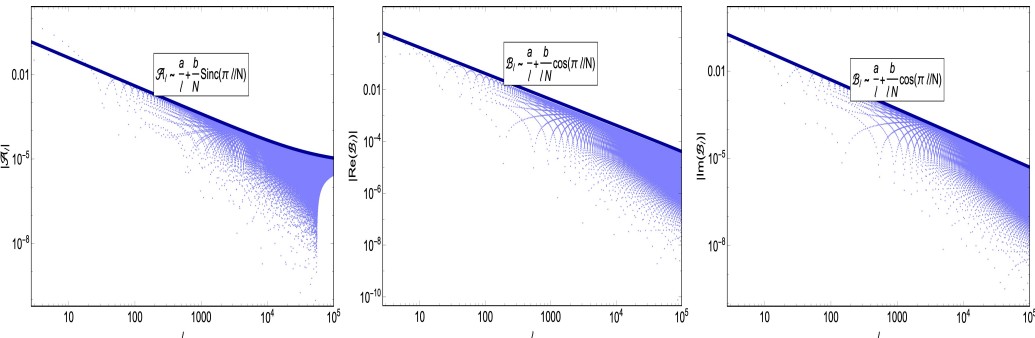

Figure 5: Effective couplings $\mathcal{A}_\ell$ (top panel) and $\mathcal{B}_\ell$ (real part in the central panel, imaginary part in the right one) as a function of distance in a non-local region, with $J = 1$, $\gamma = 0.8$, $h_1 = 10$, $h_2 = 0.5$, $\alpha = 0.5$, $T = 0.6$, and $N = 1000001$. The dots indicate the absolute value of the corresponding coupling, while the continuous lines show the decaying envelope, which decays as $\ell^{-1}$ up to finite size corrections.

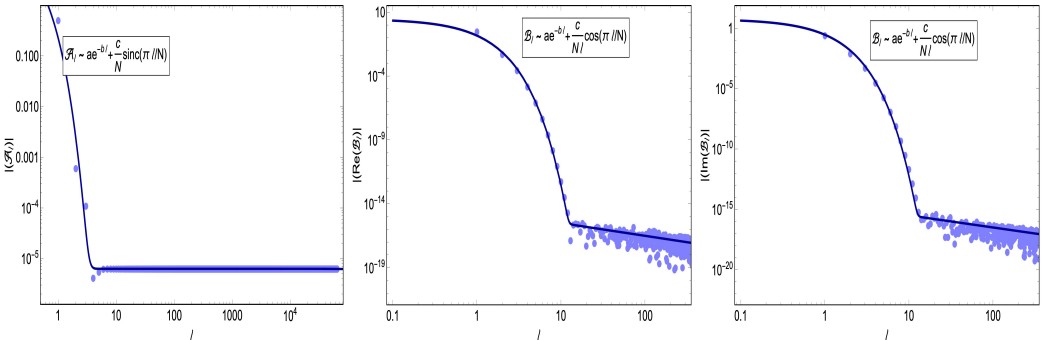

Figure 6: Effective couplings $\mathcal{A}_\ell$ (left panel) and $\mathcal{B}_\ell$ (real part in the central panel, imaginary part in the right one) as a function of distance in a local region, with $J = 1$, $\gamma = 0.8$, $h_1 = 10$, $h_2 = 0.5$, $\alpha = 0.5$, $T = 0.25$, and $N = 1000001$. The dots indicate the absolute value of the corresponding coupling, while the continuous lines show in this case an exponential decay up to finite size corrections.

We can see that all the prefactors in this asymptotic series are proportional to odd derivatives of $a_k$. However, it can be verified that this function is symmetric both around $k = 0$ and $k = \pi$, and thus its odd derivatives at these points vanish. Therefore, this series shows that the couplings must decay faster than any polynomial of $\ell^{-1}$, as can be seen in Fig. 6, where we can see an exponential decay of the couplings. This faster than algebraic decay of the function $f(\ell)$ is to be seen as a precise definition of "locality" in the present context.

## 6  Influence on the dynamics

A natural question to ask is how these features impact the evolution of observables towards the (synchronized) steady-state regime. It was previously observed that the presence of a limited number of Floquet resonances in finite-size systems has a major influence on the dynamics, leading to long-lived temporal fluctuations [18]. Similarly, in Ref. [38] open driven systems are considered where analogous regions to the ones in Fig. 1 are defined by the quick or slow spatial decay of correlation functions.

To explore this, suppose we prepare the system at an initial time in the ground state $|\text{GS}\rangle$ of a static Hamiltonian (with e.g. $h(t) = h_1$) and analyse its stroboscopic properties after introducing driving. For a given observable $\mathcal{O}$, the stroboscopic time evolution is given by $\langle \mathcal{O}(nT) \rangle \equiv \langle \Psi(nT) | \mathcal{O} | \Psi(nT) \rangle$, where $|\Psi(nT)\rangle = U(nT, 0)|\text{GS}\rangle$, with $n \in \mathbb{N}$. This can be separated into two contributions: a decaying part tending to zero as $n \to \infty$ and a (synchronized) steady-state [11] determined by the diagonal ensemble in the basis of Floquet eigenstates.

Let us work out these expressions explicitly for different observables. Consider the following operators, which can be written in pseudo-spin representation as

$$\frac{1}{N} \sum_{j=1}^{N} S_j^z = \frac{1}{2} - \frac{1}{N} \sum_{k>0} \left( \mathbb{1}_k + \sigma_k^z \right),$$

$$\frac{1}{N} \sum_{j=1}^{N} S_j^x S_{j+1}^x = -\frac{1}{2N} \sum_{k>0} \left( \cos(k)\left( \mathbb{1}_k + \sigma_k^z \right) - \sin(k)\sigma_k^y \right). \tag{31}$$

Each of these operators can be expanded as $\mathcal{O} = \sum_{k>0} \mathcal{O}_k$. Similarly, in all performed calculations the system is initially prepared in the ground state $|\text{GS}\rangle$ of an XY Hamiltonian (for instance with $h = h_1$), which can also be expanded in momentum and Floquet modes as

$$|\text{GS}_k\rangle = \sum_{\sigma \in \{-,+\}} \left| \Phi_{k,\sigma}^{[0,T]}(0) \right\rangle \left\langle \Phi_{k,\sigma}^{[0,T]}(0) \middle| \text{GS}_k \right\rangle. \tag{32}$$

Following the previous sections, the time evolution can be explicitly calculated. Taking $U_k(t = nT + \delta t, 0) = U_k(\delta t, 0)[U_k(T, 0)]^n$ with $\delta t \in [0, T)$ leads to

$$\begin{aligned}
|\Psi_k(t)\rangle &\equiv U_k(t, 0)|\text{GS}_k\rangle \\
&= \sum_{\sigma \in \{-,+\}} U_k(\delta t, 0)[U_k(T, 0)]^n \left| \Phi_{k,\sigma}^{[0,T]}(0) \right\rangle \left\langle \Phi_{k,\sigma}^{[0,T]}(0) \middle| \text{GS}_k^{(1)} \right\rangle \\
&= \sum_{\sigma \in \{-,+\}} e^{-in\sigma \epsilon_T(k)T} U_k(\delta t, 0) \left| \Phi_{k,\sigma}^{[0,T]}(0) \right\rangle \left\langle \Phi_{k,\sigma}^{[0,T]}(0) \middle| \text{GS}_k^{(1)} \right\rangle.
\end{aligned}$$

The dynamical expectation values of the above operators are then given by

$$\begin{aligned}
\langle \mathcal{O}(t) \rangle &= \sum_{\sigma, \sigma' \in \{-,+\}} \sum_{k>0} e^{-in(\sigma - \sigma')\epsilon_T(k)T} \left\langle \text{GS}_k \middle| \Phi_{k,\sigma'}^{[0,T]}(0) \right\rangle \\
&\quad \times \left\langle \Phi_{k,\sigma'}^{[0,T]}(0) \middle| U_k^\dagger(\delta t, 0) \mathcal{O}_k U_k(\delta t, 0) \middle| \Phi_{k,\sigma}^{[0,T]}(0) \right\rangle \left\langle \Phi_{k,\sigma}^{[0,T]}(0) \middle| \text{GS}_k \right\rangle.
\end{aligned}$$

These can be naturally separated in a steady-state (synchronized) term depending only on $\delta t$ and a remaining time-dependent decaying term, $\langle \mathcal{O}(t) \rangle = \mathcal{O}^{(\text{ss})}(\delta t) + \mathcal{O}^{(\text{decay})}(t)$ as

$$\begin{aligned}
\mathcal{O}^{(\text{ss})}(\delta t) &= \sum_{\sigma \in \{-,+\}} \sum_{k>0} \left\langle \text{GS}_k \middle| \Phi_{k,\sigma}^{[0,T]}(0) \right\rangle \left\langle \Phi_{k,\sigma}^{[0,T]}(0) \middle| U_k^\dagger(\delta t, 0) \mathcal{O}_k U_k(\delta t, 0) \middle| \Phi_{k,\sigma}^{[0,T]}(0) \right\rangle \\
&\quad \times \left\langle \Phi_{k,\sigma}^{[0,T]}(0) \middle| \text{GS}_k \right\rangle, \\
\mathcal{O}^{(\text{decay})}(t) &= \sum_{\sigma \in \{-,+\}} \sum_{k>0} e^{-2in\sigma \epsilon_T(k)T} \left\langle \text{GS}_k \middle| \Phi_{k,-\sigma}^{[0,T]}(0) \right\rangle \\
&\quad \times \left\langle \Phi_{k,-\sigma}^{[0,T]}(0) \middle| U_k^\dagger(\delta t, 0) \mathcal{O}_k U_k(\delta t, 0) \middle| \Phi_{k,\sigma}^{[0,T]}(0) \right\rangle \left\langle \Phi_{k,\sigma}^{[0,T]}(0) \middle| \text{GS}_k \right\rangle.
\end{aligned}$$

Let us analyse each of these distinct behaviors.

## 6.1 Decay to the synchronized state

The dynamical behaviour of observables can be shown to differ greatly depending on the different phases of the system (see Fig. 7). In the non-local regime, convergence towards the steady-state (synchronized) regime is robust, happening as $t^{-1/2}$. In the local regime on the other hand, convergence generally behaves as $t^{-3/2}$, although specific driving protocols also allow for a slower decay as $t^{-3/4}$ or $t^{-1/2}$. This behaviour is independent of the specific observable, and here it will be shown to follow from the dispersion relation of the quasienergies as function of momentum $k$. Provided $\mathcal{O}$ can be decomposed in momentum subsectors, in the thermodynamic limit the decaying part of the observable can be written as

$$\mathcal{O}^{(\text{decay})}(nT) = \frac{N}{2\pi} \int_0^\pi \mathrm{d}k \left[ f(k) e^{-2in\theta_T(k)} + \text{h.c.} \right], \tag{33}$$

in which $\theta_T(k) = \epsilon_T(k)T$ and $f(k) = \left\langle \text{GS}_k \middle| \Phi_{k,-}^{[0,T]}(0) \right\rangle \left\langle \Phi_{k,-}^{[0,T]}(0) \middle| \mathcal{O}_k \middle| \Phi_{k,+}^{[0,T]}(0) \right\rangle \left\langle \Phi_{k,+}^{[0,T]}(0) \middle| \text{GS}_k \right\rangle$. The latter contains both an observable-dependent part with the off-diagonal matrix elements of $\mathcal{O}_k$ and an observable-independent part containing the overlaps of the Floquet modes with the initial state.

The long-time dynamics of such expressions then follows from the stationary-phase approximation, where the dominant contributions to the integral will follow from stationary points where the phase $\theta_T(k)$ becomes extremal, i.e. $\theta_T'(k) = 0$. Each one of these points contributes to the dominant term in the long-time dynamics. First of all, it can be easily checked that the dispersion $\theta_T(k)$ always has extremal points at the boundaries of the Brillouin zone, where $\theta_T'(0) = \theta_T'(\pi) = 0$. However, the crucial observation is that, at $k = 0$ and $k = \pi$, the prefactor of these contributions also vanishes, $f(0) = f(\pi) = 0$. This behaviour is independent of the observable, only following from the overlaps in $f(k)$.

Indeed, let us look at the $k = 0, \pi$ sectors of the XY Hamiltonian. For these modes, the off-diagonal elements in Eq. (7) vanish, and the Hamiltonian is already diagonal in the basis $\{|0\rangle_k, c_k^\dagger |0\rangle_k\}_{k=0,\pi}$, reflecting the fact that these modes cannot be doubly occupied ($c_0^\dagger c_0^\dagger = c_\pi^\dagger c_{-\pi}^\dagger = 0$). In order to respect the separation in sectors of parity of $N + M$ of the Hilbert space, we only consider initial states with either a $|0\rangle_k$ or $c_k^\dagger |0\rangle_k$ wavefunction component for these momenta. The same holds for the Floquet modes, which are by construction restrained to one of these parity subspaces, and therefore one of the overlaps in $f(k)$ will always vanish at these points. From the stationary phase method, the contributions to the integral of these two stationary points can then be approximated as

$$\int_{-\infty}^\infty \mathrm{d}k \frac{1}{2} f''(k_0)(k-k_0)^2 e^{-in\theta_T''(k_0)(k-k_0)^2} \propto n^{-3/2}, \text{ for } \theta_T'(k_0) = 0 \text{ and } k_0 = 0, \pi. \tag{34}$$

Note that this does not necessitate $f'(k_0) = 0$, since this first-order contribution vanishes by the symmetry of the integral. If $k = 0$ and $k = \pi$ are the only stationary points, these are the only two contributions, both scaling as $n^{-3/2}$. This is precisely what happens in the local regime, where the dispersion remains qualitatively similar to that of the time-averaged Hamiltonian, leading to a decay of $t^{-3/2}$.

It can also easily be checked that additional stationary points arise in the non-local regime, corresponding to the avoided crossings in the Floquet spectrum. At these points, $f(k)$ is not expected to vanish, leading to contributions which can be approximated as

$$\int_{-\infty}^\infty \mathrm{d}k f(k_0) e^{-in\theta_T''(k_0)(k-k_0)^2} \propto n^{-1/2}, \text{ for } \theta_T'(k_0) = 0 \text{ and } k_0 \neq 0, \pi. \tag{35}$$

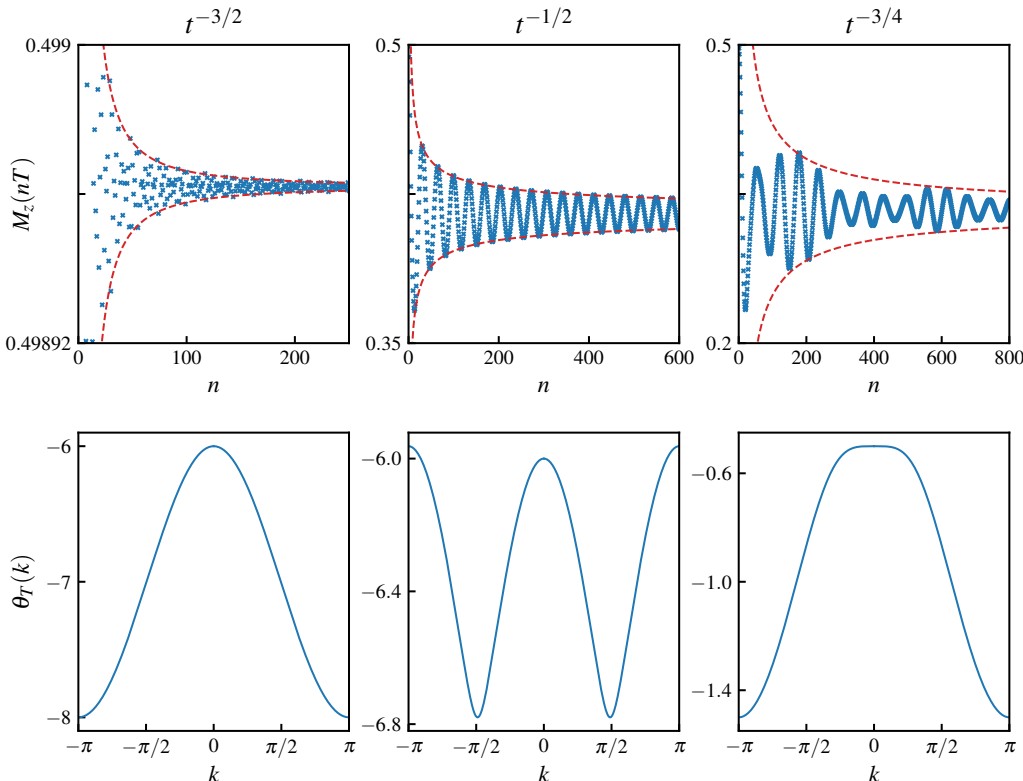

Figure 7: Possible different behaviours of the decay to the steady-state (synchronized) regime starting from the ground state of $H_{av}$ at $t = 0$. The stroboscopic time evolution of the magnetization $M_z(nT) = \langle S^z(nT) \rangle / N$ is presented in the top row, where the bottom row presents the corresponding dispersion $\theta_T(k)$. Column (i) corresponds to $t^{-3/2}$ decay in the local region, column (ii) to $t^{-1/2}$ decay in the non-local region, and column (iii) to $t^{-3/4}$ decay in the local region when $\theta_T''(0) = 0$. The stationary points at $k = 0$ and $k = \pm \pi$ can be clearly observed in all columns, whereas additional stationary points are introduced in the second column due to the presence of avoided crossings. The system parameters correspond to $N = 10001$ and (i) $\alpha = 1/2$, $\gamma = 0.5$, $J = 1$, $h_1 = 2$, $h_2 = 12$ and $T = 0.25$ (ii) $\alpha = 1/2$, $\gamma = 0.5$, $J = 1$, $h_1 = 2$, $h_2 = 12$ and $T = 0.45$ (iii) $\alpha = 1/2$, $\gamma = 0.70711$, $J = 1$, $h_1 = 0.6$, $h_2 = 0.8$ and $T = 0.1$.

Once such an additional stationary point is present, as is the case in the non-local regime, this $n^{-1/2}$ term will dominate the time evolution, leading to a slow decay of $t^{-1/2}$.

Within the local regime, there is still the possibility of obtaining a $t^{-1/2}$ decay, albeit with smaller deviations from the steady state, if $h_{av} < J(1 - \gamma^2)$, when the dispersion has one extremum in the $(0, \pi)$ interval. Remarkably, it is also possible to obtain a $t^{-3/4}$ decay by fine-tuning the system parameters. In all derivations so far, it was assumed that $\theta_T''(k_0) \neq 0$ at $k_0 = 0, \pi$. However, in some specific cases the second and the third order derivatives vanish at either $k = 0$ or $k = \pi$ (an explicit expression can be straightforwardly obtained, although it is rather unwieldy). Then, the contributions from these two stationary points are given by

$$\int_{-\infty}^{\infty} dk \frac{1}{2} f''(k_0)(k - k_0)^2 e^{-in\theta_T''''(k_0)(k-k_0)^4/12} \propto n^{-3/4},$$
$$\text{for } \theta_T'(k_0) = \theta_T''(k_0) = \theta_T'''(k_0) = 0 \text{ and } k_0 = 0, \pi. \tag{36}$$

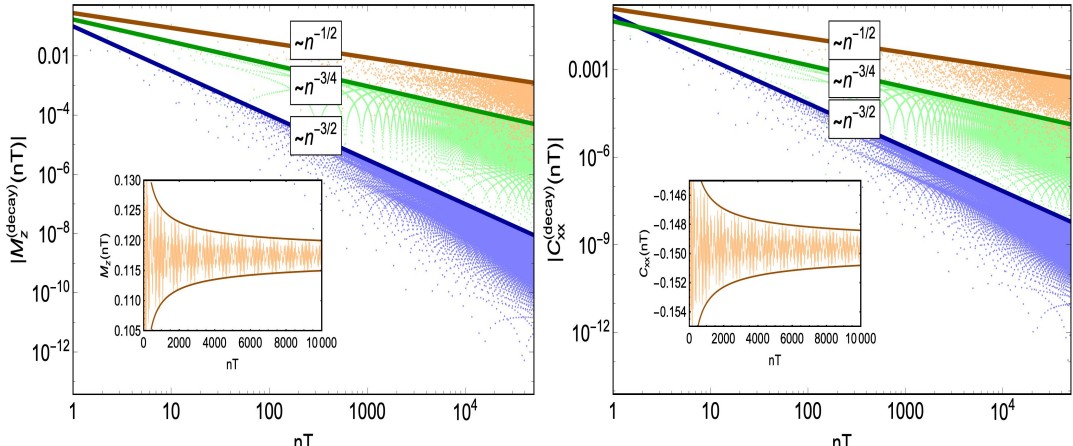

Figure 8: Stroboscopic time evolution of the decaying part of the total magnetization $M_z = \sum_{i=1}^{N} S_i^z / N$ (top panel) and nearest neighbour correlator $C_{xx} = \sum_{i=1}^{N} S_j^x S_{j+1}^x / N$ (lower panel) for $N = 100001$, $J = 1$, $t_0 = 0$ and the following choice of parameters. Orange dots: $\alpha = 0.5$, $\gamma = 0.8$, $h_1 = 0.2$, $h_2 = 1.2$, and $T = 10$; green dots: $\alpha = 0.5$, $\gamma = 0.550961$, $h_1 = 0.2$, $h_2 = 1.2$, and $T = 0.75$; blue dots: $\alpha = 0.5$, $\gamma = 0.8$, $h_1 = 2.0$, $h_2 = 0.5$, and $T = 0.75$. The value of $\gamma$ in the second case (green dots) results from imposing that the second derivative of the quasi-energy is zero at $k = 0$. In all cases we plot the absolute value of the observable so that the algebraic decay (shown by solid lines with corresponding matching colors) can be appreciated in a log-log scale. In the insets we show the total expectation value for each observable in the regime corresponding to a decay $\sim n^{-1/2}$.

If no other stationary points are present, the $n^{-3/4}$ terms will dominate the decay. These three decay regimes are illustrated in Fig. 8.

This shows that in the non-local regimes, the system generally exhibits the slowest decaying exponent. Furthermore, it exhibits persistent oscillations, the frequency of which corresponds to the quasi-energy difference between the resonant states in the avoided crossings. These can be interpreted as Rabi oscillations where the resonant states repeatedly emit and reabsorb energy to/from the driving. This change of behaviour as we transition between local to non-local regions can also be connected to crossovers between area-law to volume-law scaling in the entanglement entropy of the steady state, which was similarly shown to be reflected in dynamical phase transitions [39].

## 6.2 Synchronized state

In previous studies, it has been shown that the synchronized state can be described by a Gibbs ensemble. In most instances the temperature associated to that description turns out to be infinite [14–16]. In our case, however, due to the restriction of Floquet resonances to the single-particle sector, the system does not heat up to an infinite-temperature state, but is instead described by a non-trivial Periodic Generalized Gibbs Ensemble (PGGE) [12]. This implies that the resulting steady-state values of physically relevant observables will be highly dependent on the driving protocol, and possibly prone to display signals of phase transitions. This behaviour is indeed observed in Fig. 9, where the steady-state observables present non-analyticities precisely at the edges of local and non-local regimes. Furthermore, non-analyticities can be found well within the non-local regions, as indicated by the vertical pale green lines in the same figure. These correspond to the change in the number of resonant modes and it can be shown

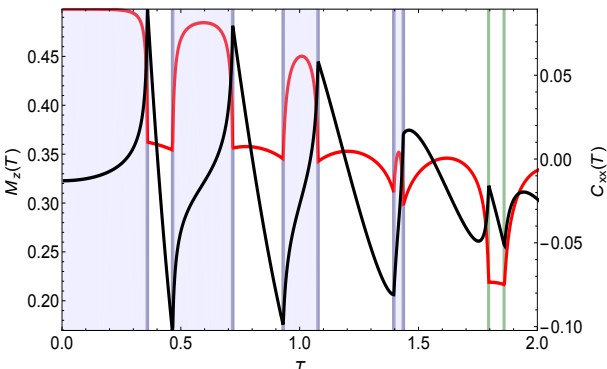

Figure 9: Synchronous value of the total magnetization $M_z = \sum_{i=1}^{N} S_i^z / N$ (red line) and correlator $C_{xx} = \sum_{i=1}^{N} S_j^x S_{j+1}^x / N$ (black line) along the dotted line $h_1 = 15$ in Fig. 1. The transitions from local to non-local regions are accompanied by the presence of non-analyticities. The vertical green lines correspond to non-analyticities inside non-local regions.

that their location in the parameter space also obeys Eqs. (27).

Note that, even though the dynamics are independent of the unfolding of the spectrum, the match between the transitions present in the Floquet Hamiltonian and those in the expectation values of the observables further justifies our choice of unfolding. This approach also circumvents the use of an extended Hilbert space [50] for which exact solutions are out of reach. Finally, from Fig. 9 we can see that by choosing certain driving parameters one can target specific combinations of values for the magnetization and correlation functions as a means of engineering systems with particular properties.

# 7 Conclusions

In this work, we considered an anisotropic XY spin chain driven by a periodically-quenched external transverse field. The algebraic structure of the theory, together with the quench-based choice of protocol, have allowed us to obtain an exact expression for the Floquet Hamiltonian and its eigenstates. While the choice of Floquet Hamiltonian for a fixed $t_0$ is unique only up to discrete gauge choices, we have shown that a simple unfolding procedure allows to clearly highlight the different non-equilibrium phases. We have established the presence of regions with local and non-local interactions depending on the driving parameters, and present analytic expressions for the phase diagram as determined by the structure of the avoided crossings of the quasi-energies representing resonances. In the local regions this Floquet Hamiltonian is well approximated by the time-averaged one, whereas the non-local region is characterized by the presence of resonances in the Floquet single-particle spectrum, necessitating long-range interactions. For these systems, there exists a maximal value of the driving period above which the interactions remain non-local. The different phases can be observed in both the dynamics and the steady-state observables. Non-local regions are distinguished by a slow decay combined with persistent oscillations originating from Floquet resonances, whereas in local regions a quicker decay to the synchronized state can be expected, though this depends on the parameters of the driving. In future work, we will return to the questions of how more general interactions influence these effects, and how they manifest themselves for other types of observables such as longer-range and/or dynamical correlations.

## Acknowledgements

J.-S. C. acknowledges support from the Netherlands Organization for Scientific Research (NWO), and from the European Research Council under ERC Advanced grant 743032 DYNAMINT. This work is part of the Delta ITP consortium, a program of the Netherlands Organisation for Scientific Research (NWO) that is funded by the Dutch Ministry of Education, Culture and Science (OCW). P.W.C. acknowledges support from a Ph.D. fellowship and a travel grant for a long stay abroad at the University of Amsterdam from the Research Foundation Flanders (FWO Vlaanderen). S.E.T.A. acknowledges support from the Foundation for Fundamental Research on Matter (FOM), which is part of the Netherlands Organization for Scientific Research (NWO). I. P. C. thanks the Delta institute for hospitality and financial support and also acknowledges partial financial support from funding UNAM-DGAPA-PAPIIT-IN106219.

## A    The antiferromagnetic XY model

We consider the spin-1/2 antiferromagnetic chain [24] described by the Hamiltonian

$$H^{XY} = J \sum_{j=1}^{N} \left[ (1+\gamma) S_j^x S_{j+1}^x + (1-\gamma) S_j^y S_{j+1}^y \right] - h \sum_{j=1}^{N} S_j^z, \tag{37}$$

where $J > 0$ is a superexchange coupling, $\gamma \in [0,1]$ is the anisotropy parameter, $h$ is an external magnetic field, $S_j^\alpha$ for $\alpha \in \{x,y,z\}$ are spin-1/2 operators and we consider periodic boundary conditions $S_{j+N}^\alpha = S_j^\alpha$. Recall that the raising and lowering spin operators are defined as $S_j^\pm = S_j^x \pm i S_j^y$. After performing a Jordan-Wigner transformation

$$
\begin{aligned}
S_j^z &= \frac{1}{2} - c_j^\dagger c_j, \\
S_j^- &= (-1)^j e^{i\pi \sum_{\ell < j} c_\ell^\dagger c_\ell} c_j^\dagger, \\
S_j^+ &= (-1)^j e^{i\pi \sum_{\ell < j} c_\ell^\dagger c_\ell} c_j,
\end{aligned}
\tag{38}
$$

where $c$ and $c^\dagger$ are operators for spinless fermions with the usual anticommutation relations $\{c_j, c_{j'}^\dagger\} = \delta_{jj'}$, we obtain the following Hamiltonian

$$
\begin{aligned}
H^{XY} = &-\frac{J}{2} \sum_{j=1}^{N-1} \left[ c_j^\dagger c_{j+1} + \gamma c_j^\dagger c_{j+1}^\dagger + \text{h.c.} \right] + h \sum_{j=1}^{N} c_j^\dagger c_j - \frac{hN}{2} \\
&+ \frac{J}{2} (-1)^N P \left[ c_N^\dagger c_1 + \gamma c_N^\dagger c_1^\dagger + \text{h.c.} \right],
\end{aligned}
\tag{39}
$$

where $P \equiv e^{i\pi \sum_{\ell=1}^{N} c_\ell^\dagger c_\ell} \equiv e^{i\pi M}$ is the operator measuring the parity of the total number of fermions, which is related to the total magnetization along $\hat{z}$. The last term in Eq. 39 corresponds to the transformation of the boundary term under periodic boundary conditions. Since $[H^{XY}, P] = 0$ the Hilbert space splits into two subspaces with even or odd $N+M$. Moreover, as the periodic boundary conditions for the spin operators are, for $c$-operators, transformed into the condition $(-1)^N e^{i\pi \sum_{\ell=1}^{N} c_\ell^\dagger c_\ell} c_{N+1} = c_1$, this implies that in the $N+M$ even sector we have $c_{N+1} = c_1$, while for the odd sector we obtain, in turn, $c_{N+1} = -c_1$. All in all, the Hamiltonian takes the form

$$H^\pm = -\frac{J}{2} \sum_{j=1}^{N} \left[ c_j^\dagger c_{j+1} + \gamma c_j^\dagger c_{j+1}^\dagger + \text{h.c.} \right] + h \sum_{j=1}^{N} c_j^\dagger c_j - \frac{hN}{2}, \tag{40}$$

where $H^+$ obeys periodic boundary conditions while $H^-$ obeys anti-periodic ones. Finally, we write the $c$-operators in Fourier modes

$$c_j = \frac{1}{\sqrt{N}} \sum_k e^{ikj} c_k, \tag{41}$$

where $k$ takes values in a Brillouin Zone that depends on $N$ and $M$, as shown in Table 1.

Table 1: Brillouin zones for different parities of $N$ and $M$.

| $N+M$ | $N$ | $M$ | Brillouin Zone | Boundary Condition |
|---|---|---|---|---|
| even | even | even | $BZ_1 = \{-\pi + \frac{2\pi}{N}, \ldots, 0, \ldots, \pi\}$ | PBC |
| even | odd | odd | $BZ_2 = \{-\pi + \frac{2\pi}{N}, \ldots, 0, \ldots, \pi - \frac{2\pi}{N}\}$ | PBC |
| odd | even | odd | $BZ_3 = \{-\pi + \frac{\pi}{N}, \ldots, -\frac{\pi}{N}, \frac{\pi}{N}, \ldots, \pi - \frac{\pi}{N}\}$ | APBC |
| odd | odd | even | $BZ_4 = \{-\pi + \frac{2\pi}{N}, \ldots, -\frac{\pi}{N}, \frac{\pi}{N}, \ldots, \pi\}$ | APBC |

For a given Brillouin zone, for instance $BZ_3$, the Hamiltonian in the reciprocal space reads

$$
\begin{aligned}
H^{XY} = -J \sum_{k>0} \begin{pmatrix} c_k^\dagger & c_{-k} \end{pmatrix} \begin{pmatrix} \cos(k) - \frac{h}{J} & i\gamma \sin(k) \\ -i\gamma \sin(k) & -(\cos(k) - \frac{h}{J}) \end{pmatrix} \begin{pmatrix} c_k \\ c_{-k}^\dagger \end{pmatrix} \\
- \frac{hN}{2} - J \sum_{k>0} \left( \cos(k) - \frac{h}{J} \right).
\end{aligned}
\tag{42}
$$

This Hamiltonian can then be diagonalised by a Bogoliubov transformation

$$
\begin{pmatrix} c_k \\ c_{-k}^\dagger \end{pmatrix} = \begin{pmatrix} \cos(\theta_k/2) & -i\sin(\theta_k/2) \\ -i\sin(\theta_k/2) & \cos(\theta_k/2) \end{pmatrix} \begin{pmatrix} \eta_k \\ \eta_{-k}^\dagger \end{pmatrix},
\tag{43}
$$

with a Bogoliubov angle defined by $\sin(\theta_k) = -\lambda_k/\sqrt{\xi_k^2 + \lambda_k^2}$ and $\cos(\theta_k) = -\xi_k/\sqrt{\xi_k^2 + \lambda_k^2}$, where $\lambda_k = \gamma \sin(k)$ and $\xi_k = \cos(k) - h/J$. In terms of Bogoliubov fermions, the Hamiltonian becomes

$$H^{XY} = \sum_k E(k) \eta_k^\dagger \eta_k - hN/2 - \sum_{k>0} (J\cos(k) - h + E(k)), \tag{44}$$

with excitation spectrum

$$E(k) = J\sqrt{\lambda_k^2 + \xi_k^2}. \tag{45}$$

The model undergoes a quantum phase transition when $h = J$, with an antiferromagnetic ground state for $h < J$, and a paramagnetic one for $h > J$. The dispersion relation (45) has a minimum at $k = 0$ when $h \geq J(1 - \gamma^2)$, and at $k = \arccos(h/J(1 - \gamma^2))$ when $h < J(1 - \gamma^2)$. In the latter case, $k = 0$ becomes a local maximum. In both situations, there is a maximum for $k = \pi$.

Another approach to the problem comes from noticing that the Hamiltonian (42) allows a pseudospin 1/2 representation since it connects the Fock vacuum $|0\rangle$ to the state $c_k^\dagger c_{-k}^\dagger |0\rangle$ for each $k$. We thus introduce the following representation

$$c_k^\dagger c_{-k}^\dagger |0\rangle_k \to \begin{pmatrix} 1 \\ 0 \end{pmatrix}_k \equiv |\mathbf{1}\rangle_k, \qquad |0\rangle_k \to \begin{pmatrix} 0 \\ 1 \end{pmatrix}_k \equiv |\mathbf{2}\rangle_k. \tag{46}$$

Thus operators acting on this space, and expressible in terms of sum of operators acting on $k$-labeled subspaces (e.g. $O = \sum_{k>0} O_k$) can be written, in this representation, as sums of

$2 \times 2$ matrices whose entries are $(O_k)_{ij} = \langle \mathbf{i} | O_k | \mathbf{j} \rangle$ for $i, j \in \{1, 2\}$. For the Hamiltonian (42) this allows to write it as

$$H^{XY} = \sum_{k>0} H_k - \frac{hN}{2} - J \sum_{k>0} \xi_k \,, \tag{47}$$

where $H_k = E(k) \mathbf{n}_k \cdot \sigma_k$ with $\sigma_k = (\sigma_k^x, \sigma_k^y, \sigma_k^z)$ being Pauli matrices, and $\mathbf{n}_k = (0, -\sin(\theta_k), \cos(\theta_k))$. This representation is related to Eq. (44) through a particle-hole transformation of the negative momentum Bogoliubov modes. Thus, the diagonalized Hamiltonian presents two energy branches $E_\sigma(k) = \sigma E(k)$ for $\sigma \in \{-1, 1\}$ with eigenvectors

$$\left| E_{\sigma,k}^{(i)} \right\rangle = a_\sigma(k) \begin{pmatrix} 1 \\ \frac{E_\sigma(k) - \xi_k}{i\lambda_k} \end{pmatrix}, a_\sigma(k) = \frac{1}{\sqrt{1 + \frac{[E_\sigma(k) - \xi_k]^2}{\lambda_k^2}}} \,. \tag{48}$$

The pseudospin representation is particularly useful as it allows to express time evolution as rotations on the Bloch sphere around the axis $\mathbf{n}_k$, that is,

$$U_k(t', t) = e^{-iH_k(t'-t)} = \cos[E(k)(t'-t)] \mathbb{1} - i \sin[E(k)(t'-t)] \mathbf{n}_k \cdot \sigma_k \,. \tag{49}$$

## B  Other Floquet gauge choices and continuous time evolution

In the main text we focus on the case where $t_0 = 0$ and consider mostly the stroboscopic dynamics. Suppose, however, that we start the evolution of the system at an arbitrary Floquet gauge $t_0$ and we want to evolve it up to a time $t_0 + \delta t$, $\delta t \in [0, T]$. Then the evolution operator in a given $k$ sector is given by

$$U_k(t_0 + \delta t, t_0) = \begin{cases} e^{-iH_k^{(1)}\delta t}, & \text{if } \delta t \in [0, \alpha T - t_0] \,, \\ e^{-iH_k^{(2)}(t_0 + \delta t - \alpha T)} e^{-iH_k^{(1)}(\alpha T - t_0)}, & \text{if } \delta t \in [\alpha T - t_0, T - t_0] \,, \\ e^{-iH_k^{(1)}(t_0 + \delta t - T)} e^{-iH_k^{(2)}(1-\alpha)T} e^{-iH_k^{(1)}(\alpha T - t_0)}, & \text{if } \delta t \in [T - t_0, T] \,, \end{cases} \tag{50}$$

where $H_k^{(i)}$ is the Hamiltonian with field $h_i$, with eigenvalues $E_i(k)$ and Bogoliubov angles $\theta_k^{(i)}$. Using an Euler angle representation, we will write this evolution operator as

$$\begin{aligned} U_k(t_0 + \delta t, t_0) &= u_{k,0}(\delta t) \mathbb{1}_k + i u_{k,x}(\delta t) \sigma_k^x + i u_{k,y}(\delta t) \sigma_k^y + i u_{k,z}(\delta t) \sigma_k^z \\ &= \begin{pmatrix} u_{k,0}(\delta t) + i u_{k,z}(\delta t) & u_{k,y}(\delta t) + i u_{k,x}(\delta t) \\ -u_{k,y}(\delta t) + i u_{k,x}(\delta t) & u_{k,0}(\delta t) - i u_{k,z}(\delta t) \end{pmatrix}, \end{aligned} \tag{51}$$

where the coefficients $u_{k,l}(\delta t)$ are given by

$$u_{k,0}(\delta t) = \begin{cases} \cos[E_1(k)\delta t], \\ \cos[E_1(k)(\alpha T - t_0)] \cos[E_2(k)(t_0 + \delta t - \alpha T)] \\ \quad - \sin[E_1(k)(\alpha T - t_0)] \sin[E_2(k)(t_0 + \delta t - \alpha T)] \cos \Delta_k \,, \\ \cos[E_1(k)(t_0 + \delta t - T)]\{\cos[E_1(k)(\alpha T - t_0)] \cos[E_2(k)(1-\alpha)T] \\ \quad - \sin[E_1(k)(\alpha T - t_0)] \sin[E_2(k)(1-\alpha)T] \cos \Delta_k\} \\ \quad - \sin[E_1(k)(t_0 + \delta t - T)]\{\cos[E_1(k)(\alpha T - t_0)] \sin[E_2(k)(1-\alpha)T] \cos \Delta_k \\ \quad + \sin[E_1(k)(\alpha T - t_0)] \cos[E_2(k)(1-\alpha)T]\} \,, \end{cases} \tag{52}$$

$$
u_{k,x}(\delta t) = \begin{cases}
0\,, \\
\sin[E_1(k)(\alpha T - t_0)]\sin[E_2(k)(t_0 + \delta t - \alpha T)]\sin(\Delta_k)\,, \\
\sin[E_1(k)(\alpha T - t_0)]\cos[E_1(k)(t_0 + \delta t - T)]\sin[E_2(k)(1 - \alpha)T]\sin(\Delta_k) \\
\quad - \sin[E_1(k)(t_0 + \delta t - T)]\cos[E_1(k)(\alpha T - t_0)]\sin[E_2(k)(1 - \alpha)T]\sin(\Delta_k)\,,
\end{cases}
\tag{53}
$$

$$
u_{k,y}(\delta t) = \begin{cases}
\sin[E_1(k)\delta t]\sin\!\left(\theta_k^{(1)}\right)\,, \\
\cos[E_1(k)(\alpha T - t_0)]\sin[E_2(k)(t_0 + \delta t - \alpha T)]\sin\!\left(\theta_k^{(2)}\right) \\
\quad + \sin[E_1(k)(\alpha T - t_0)]\cos[E_2(k)(t_0 + \delta t - \alpha T)]\sin\!\left(\theta_k^{(1)}\right)\,, \\
\sin[E_1(k)(t_0 + \delta t - T)]\{\cos[E_1(k)(\alpha T - t_0)]\cos[E_2(k)(1 - \alpha)T]\}\sin\!\left(\theta_k^{(1)}\right) \\
\quad + \sin[E_1(k)(t_0 + \delta t - T)]\sin[E_1(k)(\alpha T - t_0)]\sin[E_2(k)(1 - \alpha)T]\sin\!\left(\Delta_k - \theta_k^{(1)}\right) \\
\quad + \cos[E_1(k)(t_0 + \delta t - T)]\{\cos[E_1(k)(\alpha T - t_0)]\sin[E_2(k)(1 - \alpha)T]\sin\!\left(\theta_k^{(2)}\right) \\
\quad + \sin[E_1(k)(\alpha T - t_0)]\cos[E_2(k)(1 - \alpha)T]\sin\!\left(\theta_k^{(1)}\right)\}\,,
\end{cases}
\tag{54}
$$

$$
u_{k,z}(\delta t) = \begin{cases}
-\sin[E_1(k)\delta t]\cos\!\left(\theta_k^{(1)}\right)\,, \\
-\cos[E_1(k)(\alpha T - t_0)]\sin[E_2(k)(t_0 + \delta t - \alpha T)]\cos\!\left(\theta_k^{(2)}\right) \\
\quad -\sin[E_1(k)(\alpha T - t_0)]\cos[E_2(k)(t_0 + \delta t - \alpha T)]\cos\!\left(\theta_k^{(1)}\right)\,, \\
-\cos[E_1(k)(t_0 + \delta t - T)]\{\cos[E_1(k)(\alpha T - t_0)]\sin[E_2(k)(1 - \alpha)T]\cos\!\left(\theta_k^{(2)}\right) \\
\quad + \sin[E_1(k)(\alpha T - t_0)]\cos[E_2(k)(1 - \alpha)T]\cos\!\left(\theta_k^{(1)}\right)\} \\
\quad -\sin[E_1(k)(t_0 + \delta t - T)]\{\cos[E_1(k)(\alpha T - t_0)]\cos[E_2(k)(1 - \alpha)T]\}\cos\!\left(\theta_k^{(1)}\right) \\
\quad + \sin[E_1(k)(t_0 + \delta t - T)]\sin[E_1(k)(\alpha T - t_0)]\sin[E_2(k)(1 - \alpha)T]\cos\!\left(\Delta_k - \theta_k^{(1)}\right)\,,
\end{cases}
\tag{55}
$$

with $\Delta_k = \theta_k^{(2)} - \theta_k^{(1)}$. The coefficients are given in a piecewise manner for $\delta t$ in the three intervals $[0, \alpha T - t_0]$, $[\alpha T - t_0, T - t_0]$, and $[T - t_0, T]$, respectively. This result will allow us to do any arbitrary time evolution and to obtain the Floquet modes by diagonalising it for $\delta t = T$. We denote the Euler parameters of the stroboscopic time evolution operator $U_k(t_0 + T, t_0)$ as $u_{k,p} \equiv u_{k,p}(\delta t = T)$, $p \in \{0, x, y, z\}$, whose expressions turn out to be

$$
\begin{aligned}
u_{k,0} &= \cos[E_1(k)t_0]\{\cos[E_1(k)(\alpha T - t_0)]\cos[E_2(k)(1 - \alpha)T] \\
&\quad - \sin[E_1(k)(\alpha T - t_0)]\sin[E_2(k)(1 - \alpha)T]\cos\Delta_k\} \\
&\quad - \sin[E_1(k)t_0]\{\cos[E_1(k)(\alpha T - t_0)]\sin[E_2(k)(1 - \alpha)T]\cos\Delta_k \\
&\quad + \sin[E_1(k)(\alpha T - t_0)]\cos[E_2(k)(1 - \alpha)T]\}\,, \\
u_{k,x} &= \sin[E_1(k)(\alpha T - t_0)]\cos[E_1(k)t_0]\sin[E_2(k)(1 - \alpha)T]\sin(\Delta_k) \\
&\quad - \sin[E_1(k)t_0]\cos[E_1(k)(\alpha T - t_0)]\sin[E_2(k)(1 - \alpha)T]\sin(\Delta_k)\,, \\
u_{k,y} &= \sin[E_1(k)t_0]\cos[E_1(k)(\alpha T - t_0)]\cos[E_2(k)(1 - \alpha)T]\sin\!\left(\theta_k^{(1)}\right) \\
&\quad + \sin[E_1(k)t_0]\sin[E_1(k)(\alpha T - t_0)]\sin[E_2(k)(1 - \alpha)T]\sin\!\left(\Delta_k - \theta_k^{(1)}\right) \\
&\quad + \cos[E_1(k)t_0]\{\cos[E_1(k)(\alpha T - t_0)]\sin[E_2(k)(1 - \alpha)T]\sin\!\left(\theta_k^{(2)}\right) \\
&\quad + \sin[E_1(k)(\alpha T - t_0)]\cos[E_2(k)(1 - \alpha)T]\sin\!\left(\theta_k^{(1)}\right)\}\,,
\end{aligned}
$$

$$u_{k,z} = -\cos[E_1(k)t_0]\{\cos[E_1(k)(\alpha T - t_0)]\sin[E_2(k)(1-\alpha)T]\cos\left(\theta_k^{(2)}\right)$$
$$+ \sin[E_1(k)(\alpha T - t_0)]\cos[E_2(k)(1-\alpha)T]\cos\left(\theta_k^{(1)}\right)\}$$
$$- \sin[E_1(k)t_0]\{\cos[E_1(k)(\alpha T - t_0)]\cos[E_2(k)(1-\alpha)T]\}\cos\left(\theta_k^{(1)}\right)$$
$$+ \sin[E_1(k)t_0]\sin[E_1(k)(\alpha T - t_0)]\sin[E_2(k)(1-\alpha)T]\cos\left(\Delta_k - \theta_k^{(1)}\right).$$

## C   Finite-size corrections

To derive expressions describing the finite size effects, an easy way to obtain the correction function $g(\ell)$ is to replace the boundaries of the integral by the lowest and highest momenta allowed by the quantization:

$$\mathcal{A}_\ell \approx \frac{1}{\pi}\,\text{Re}\int_{\pi/N}^{\pi-\pi/N} a_k e^{ik\ell}\mathrm{d}k = \frac{1}{\pi\ell}(a_{\pi-\pi/N}\sin((\pi-\pi/N)\ell) - a_{\pi/N}\sin((\pi/N)\ell) + O(\ell^{-2})$$
$$= -\frac{(-1)^\ell a_\pi + a_0}{N}\text{sinc}(\pi\ell/N) + O(\ell^{-2}) + O(N^{-2}),$$
(56)

with $\text{sinc}(x) = \sin(x)/x$. This finite size correction is the same in both the local and non-local regime. The same analysis can be done for $\mathcal{B}_\ell$ yielding the same asymptotic behaviour in both the local and non-local regimes. The finite size corrections however are slightly different, and are given by

$$g_{\mathcal{B}}(\ell) = -\frac{\cos(\pi\ell/N)}{\ell N}\left[\left((-1)^\ell \frac{\mathrm{d}b_k}{\mathrm{d}k}\bigg|_\pi + \frac{\mathrm{d}b_k}{\mathrm{d}k}\bigg|_0\right) + i\left((-1)^\ell \frac{\mathrm{d}c_k}{\mathrm{d}k}\bigg|_\pi + \frac{\mathrm{d}c_k}{\mathrm{d}k}\bigg|_0\right)\right],$$
(57)

where

$$b_k = \epsilon_T(k)u_{k,y}\Big/\sqrt{1-u_{k,0}^2}, \quad c_k = \epsilon_T(k)u_{k,x}\Big/\sqrt{1-u_{k,0}^2}.$$
(58)

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
