# Peer review of "Out-of-equilibrium phase transitions induced by Floquet resonances in a periodically quench-driven XY spin chain"

_SciPost Physics Core, doi:SciPost Phys. Core 3, 001 (2020)_

## Round 1 · Referee Report · Anonymous (Referee 1) · 2020-6-19

Report

In this work the authors study a periodically driven XY model and its dynamics as well as long-time steady states via exact analytical solutions. The central element of their approach is an exact analytical derivation of the Floquet Hamiltonian, which, in general, is a difficult task. Here, it can be achieved by mapping the XY model to a system of free fermions whose properties can be approached via analytical methods. The authors study in detail this Floquet Hamiltonian and they find that a key aspect to understand the system's properties in the absense or presence of Floquet resonances.

This paper is well written and straightforward to follow also for non-experts. The results appear sound and timely aligning well with the recent efforts of the community in the context of Floquet physics.

While the model is rather specific (exactly solvable and therefore potentially different concerning its long-time dynamics compared to a generic quantum many-body system which is expected to heat up indefinitely), the presented results such as the schemes for unfolding of Floquet spectra and the analysis of the Floquet generated long-range interactions due to resonances can be crucial also in a many-body interacting context.

It is also very important to emphasize that the manuscript not only provides a mathematical analysis of the Floquet Hamiltonian, but also explores the physical consequences of the observed Floquet resonances, which turn out to be crucial. They significantly influence not only the long-time dynamics of observables (leading to different decay laws), but also give rise to non-equilibrium phase transitions in the steady state.

Overall, the manuscript certainly warrants publication in SciPost.

I have just one suggestion concerning Fig. 3, where a proper colorbar is missing for the presented plots. I would suggest to add that in a revised version.

Requested changes

I have just one suggestion concerning Fig. 3, where a proper colorbar is missing for the presented plots. I would suggest to add that in a revised version.

---

## Editorial Decision

published